# Comparative Studies of CPEs Modified with Distinctive Metal Nanoparticle-Decorated Electroactive Polyimide for the Detection of UA

**DOI:** 10.3390/polym13020252

**Published:** 2021-01-13

**Authors:** Aamna Bibi, Sheng-Chieh Hsu, Wei-Fu Ji, Yi-Chi Cho, Karen S. Santiago, Jui-Ming Yeh

**Affiliations:** 1Department of Chemistry and Center for Nanotechnology, Chung Yuan Christian University, Chung Li District, Tao-Yuan City 32023, Taiwan; emi2118@gmail.com (A.B.); trombone0410@gmail.com (S.-C.H.); asd790508@hotmail.com (W.-F.J.); sdef69887@yahoo.com.tw (Y.-C.C.); 2Department of Chemistry, College of Science, University of Santo Tomas, España, Manila 1015, Philippines

**Keywords:** electroactive, polyimide, nanoparticles, uric acid, electrochemical sensor

## Abstract

In this present work, an electrochemical sensor was developed for the sensing of uric acid (UA). The sensor was based on a carbon paste electrode (CPE) modified with electroactive polyimide (EPI) synthesized using aniline tetramer (ACAT) decorated with reduced nanoparticles (NPs) of Au, Pt, and Ag. The initial step involved the preparation and characterization of ACAT. Subsequently, the ACAT-based EPI synthesis was performed by chemical imidization of its precursors 4,4′-(4.4′-isopropylidene-diphenoxy) bis (phthalic anhydride) BPADA and ACAT. Then, EPI was doped with distinctive particles of Ag, Pt and Au, and the doped EPIs were abbreviated as EPIS, EPIP and EPIG, respectively. Their structures were characterized by XRD, XPS, and TEM, and the electrochemical properties were determined by cyclic voltammetry and chronoamperometry. Among these evaluated sensors, EPI with Au NPs turned out the best with a sensitivity of 1.53 uA uM^−1^ UA, a low limit of detection (LOD) of 0.78 uM, and a linear detection range (LDR) of 5–50 uM UA at a low potential value of 310 mV. Additionally, differential pulse voltammetric (DPV) analysis showed that the EPIG sensor showed the best selectivity for a tertiary mixture of UA, dopamine (DA), and ascorbic acid (AA) as compared to EPIP and EPIS.

## 1. Introduction

Uric acid (UA) in the human body stems from purine metabolism and is an important marker for illness [1]. Generally, a person’s health is related to the concentration of UA in the blood (1.2–4.5 × 10^−4^ mol L^−1^) or urine (2 × 10^−3^ mol L^−1^) [2]. An abnormal concentration of UA in body fluids could be a symptom of a disease like Lesch–Nyhan syndrome [3], toxemia of pregnancy [4], or hyperuricemia [5], and along with cardiovascular diseases it may also affect the circulatory system [6]. Therefore, UA sensing with cost-effective monitoring, intended sensitivity, and accuracy in human body fluids is crucial to disease screening [7]. In general, different techniques have been acquired for UA sensing, including HPLC [8], fluorescence [9], a single line manifold [10], uricase immobilization methods [11], chemiluminescence [12], and direct electrochemical detection [13].

Among all these methods, the electrochemical technique is extensively used and has been proven to be an excellent method for the determination of UA due to its quick response, easy handling, specificity, low expenditure, and high sensitivity [14,15]. In an aqueous solution, UA undergoes oxidation at common electrodes with ease, so direct electrochemical detection leads to better sensing. Moreover, direct sensing can be extended for evaluating UA in a blood sample by the simple pin-prick method [13].

However, other potentially co-existing analytes in body fluids like AA and DA may interfere with the electrochemical detection of UA. The oxidation potential of UA, AA, and DA are too close to strongly overlap, which results in poor selectivity of electrodes. To address this concern, two techniques have been employed, i.e., polymer-modified electrodes and chemically-modified electrodes. Thus, the electrode can be modified and improved using conducting polymers (CPs), noble metal nanoparticles, organic/inorganic compounds, and ionic/polymeric film [16,17]. Conducting polymers have various properties, including electrochemical activity, electrical conductivity, mechanical strength, biocompatibility, and environmental stability, which are highly desirable for the advancement of sensing performance in analytical or bioanalytical system. Moreover, they can be easily doped by electrochemical techniques [18].

For instance, Ramanavicius et al., recently developed a quartz-crystal microbalance sensor (EQCM) based on polypyrrole imprinted by uric acid (MIP(UA)-Ppy). Experimental measurements and theoretical calculations showed that the formation of a uric acid complex with MIP(UA)-Ppy is thermodynamically more favorable than that of complexation with NIP-Ppy. However, the sensitivity of the MIP(UA)-Ppy-based sensor to uric acid in a linear dependence interval was approximately 3.5 times higher than that of a NIP-Ppy-based sensor [19]. Gryglewicz et al., developed a GCE sensor modified with nitrogen-doped reduced graphene oxide and gold nanoparticle nanocomposites to improve the simultaneous detection of DA, AA, and UA. The results showed that nitrogen doping exhibited better electrochemical performance due to the more efficient anchoring of Au NPs on the N-doped areas of the graphene structure [20]. Similarly, Ma et al., devised a system that includes a screen-printed electrode (SPE), which is used as a sensor by alternately depositing PEDOT, CS, and graphene onto the working electrode for the detection of dopamine. The system could detect DA with a detection sensitivity of 0.52 ± 0.01 uA/uM and a limit of detection of 0.29 uM in the linear range of DA concentrations from 0.05 to 70 uM [21].

Due to its superior environmental stability, ease of synthesis, low cost, and uniform conductive mechanism, PANI is a popular conjugated polymer. Doped PANI is often used in sensing applications as it shows high conductivity under acidic conditions. Generally, PANI doped with noble metal nanoparticles (NPs) enhances the electron transfer and improves the conductivity and stability. Therefore, it is used in electrochemical sensing [22,23,24] as it provides rapid and accurate sensing [25].

As a model compound of PANI, oligoanilines have been studied enormously in the past decade. Consequently, oligoaniline-derived electroactive polymers (OAEPs) have gained considerable attention [26,27,28,29] due to their good solubility, biodegradable nature, and mechanical strength [30,31,32,33]. For instance, Chao and Wang prepared OAEPs that have remarkable electrochemical and electrochromic behavior [34,35]. In 2020, Yeh et al. reported an H_2_S gas sensor based on an electroactive polymer containing aniline trimer in the main chain [36]. Additionally, Huang et al., prepared a polylactide-based OAEP for tissue engineering [37]. Some reports [38] of fullerene (C60) and carbon nanotubes (CNTs) grafted with oligoanilines were reported in the literature. Moreover, OAEPs report good thermal stability [39], electrochemical activity [40,41], biomedical applications [42], electrochromic properties [43], and electrochemical [44] and conductivity studies [45]. Additionally, in our previous studies, OAEPs have been used for electrochemical sensing of ascorbic acid [46,47,48,49,50,51,52,53,54].

To the best of our knowledge, OAEPs are generally use for electrochemical sensing of ascorbic acid [55,56] but are seldom mentioned for UA sensing. With this information in hand, we constructed an electrochemical sensor for UA sensing based upon an oligoaniline derived electroactive polyimide (EPI). Furthermore, to improve analytical performances like sensitivity and detection limit, nanosized particles of noble metals [57,58,59,60,61] were reduced on the surface of EPI. An electroactive polyimide (EPI) was prepared by a two-step process followed by a complete characterization. The as-prepared EPI was then immersed in different aqueous solutions including AgNO_3_, H_2_PtCl_6_, and HAuCl_4_ for 6 h. The presence of metal particles that were reduced on the EPI surface was examined by XRD, TEM, and XPS. The electrochemical sensors constructed for UA sensing were investigated for various studies, such as electroanalytical, kinetic, sensitivity, selectivity, and differential-pulse voltammetry.

## 2. Materials and Methods

All the chemicals purchased from Sigma-Aldrich (St. Louis, MO, USA), TCI (Tokyo, Japan) and Seechem (Seoul, Korea). All the solvents were purchased from J.T. Baker (Randor, PA, USA), Riedel-de-haën (Seelz, Germany) and Macron (Belmont, CA, USA), with 99.99% purity. HAuCl_4_•3H_2_O), H_2_PtCl_6_•6H_2_O, hydrazine, and AgNO_3_ were all from Alfa Aesar. ^1^H-NMR and LCMS were performed on a Bruker AVANCE300 spectrometer and Bruker Daltonics IT mass spectrometer model Esquire 2000 (Leipzig, Germany), respectively. An FTIR spectrometer (JASCO FT/IR-4100, Easton, PA, USA) and VoltaLab 50 (PST050) were used to perform FTIR and electrochemical experiments. The molecular weight was determined on a Waters GPC-150CV (Milford, CT, USA), and a JEOL-200FX TEM (Tokyo, Japan) was used for TEM study.

### 2.1. Synthesis of Amino-Capped Aniline Tetramer (ACAT)

As shown in Scheme I, aniline tetramer (ACAT) was prepared using a simple procedure [62]. First, diphenylamine (0.01 mol) and 4,4′-diaminodiphenyl-amine sulphate (0.01 mol) were dissolved in a tri-solvent system of DMF (100 mL), H_2_O (20 mL) and 1 N HCl (25 mL). After that, ammonium persulphate solution (2.28 g in 25 mL of 1 N HCl) was added slowly to the above-mentioned solution at 0 °C via a dropping funnel. Subsequently, after one hour of stirring at 0 °C the reaction was precipitated out in distilled water (700 mL). Finally, the vacuum filtration was used to obtain the precipitate followed by a washing with 400 mL of HCl (1 N), 100 mL of NH_4_OH (1.2 N), and distilled water. Lastly, the crude product of ACAT was dried in a vacuum at 50 °C and obtained as blue powder.

### 2.2. Synthesis of EPI

The EPI preparation was a two-step process. First, 0.52 g of 4,4′-(4.4′-isopropylidene-diphenoxy) bis (phthalic anhydride) and 0.379 g of ACAT were added to 16 g of DMAc in a two-neck-round bottom flask. The reaction mixture was stirred for 24 h at room temperature to produce electroactive polyamic acid (EPAA). Afterwards, the mixture of acetic anhydride and pyridine (0.102/0.079, *v/v*) was added to the previous solution under stirring for 1 h. Subsequently, the reaction mixture was heated at 150 °C for 3 h to obtain the EPI solution. Finally, the precipitation was carried out in methanol and the precipitate washed with hot methanol. Lastly, the EPI was dried at 60 °C in a vacuum and collected in a good yield (80%) [62].

### 2.3. Preparation of EPIS, EPIP, and EPIG

The standard procedure for the preparation of EPIS, EPIP, and EPIG was as follows: Firstly, an EPI (leucoemeraldine base) was prepared by dispersing 0.1 g of as-prepared EPI in 10 mL of 1.0 M NH_4_OH containing 1 mL of hydrazine solution. The reaction mixture was stirred for 24 h, filtered, and washed with water until the pH became neutralized. At this time, a freeze-drying technique was used at −42 °C for 24 h, and EPI (leucoemeraldine base) was collected in form of blue powder. Similarly, EPIS, EPIP, and EPIG were prepared by immersing 0.1 g EPI (leucoemeraldine base) in 19 mL of 0.1 mM AgNO_3,_ H_2_PtCl_6_•6H_2_O and HAuCl_4_•3H_2_O for 6 h, respectively. Later, the EPIS, EPIP, and EPIG powders were collected by centrifugal filtration, washed with an excess amount of distilled H_2_O and dried at 50 °C for 24 h. Finally, the as-prepared EPI decorated with distinctive metallic particles was collected as a fine blue powder. The synthetic route for EPI decorated with Ag/Pt/Au particles is shown in Scheme 1.

### 2.4. Electrochemical Cyclic Voltammetry of EPI, EPIS, EPIP, and EPIG

The redox behavior of the as-prepared EPI, EPIS, EPIP, and EPIG was determined by spin-coating these electroactive materials onto an indium titanium oxide (ITO) followed by performing a series of CV studies. Firstly, different solutions were prepared by dissolving 0.05 g of respective sample in 5 mL of DMAc under magnetic stirring for 6 h. After this, these solutions were spin-coated on an ITO and dried by heating at 120 °C for 1 h. Cyclic voltammetry (CV) of as-prepared ITOs was performed on a AutoLab PGSTAT 204 (Houten, The Netherlands).

### 2.5. Electrochemical Sensing of UA

Figure 1 shows the fabrication of a sensor for electrochemical detection of UA by using a carbon paste electrode (CPE) modified with EPIs. At first, four different samples were prepared by a thorough mixing of 0.04 g of each respective polymer (EPI, EPIS, EPIP, EPIG) with graphite powder (0.05 g) and paraffin oil (0.01) to form a homogeneous carbon paste. A portion of all these samples was packed firmly into a Teflon tube cavity, and the opposite end was inserted with a copper rod to make an electrical contact. Voltammetric studies were performed on an AutoLab PGSTAT 204 electrochemical work station in a 0.1 M phosphate buffer solution (PBS) (20 mL). Platinum foil was used as a counter electrode and Ag/AgCl (3.0 M NaCl solution) served as a reference electrode, while CPE electrodes modified with EPI, EPIS, EPIP or EPIG were employed as working electrodes.

## 3. Results

### 3.1. Characterization of ACAT

Figure 2 shows the complete characterization of ACAT. As presented in Figure 2a, the ^1^H-NMR spectrum of ACAT reveals the signal at 5.50 ppm indicating the NH_2_ protons. Moreover, the signals around 7.3–6.8 ppm show the splitting of the aromatic protons. The FTIR spectrum of ACAT in Figure 2b shows the absorption band at 3300 and 3200 cm^−1^ corresponding to the NH_2_ groups, while the vibrational bands for quinoid and benzenoid ring appears at 1600 and 1500 cm^−1^, respectively. The peak at 830 cm^−1^ may be correlated to the substitution pattern of a 1,4-disubstituted benzene ring. Moreover, Figure 2c shows the molecular weight of ACAT with a molecular ion peak appeared at 380 (*m*/*z*) [63,64].

### 3.2. Characterization of EPAA and EPI

As shown in Figure 3, FTIR was used to characterize the structure of EPAA and EPI. The FTIR spectra of EPAA 3a showed a characteristic carboxylic and amide absorption band at 3250–3500 (N–H and O–H spectra), 1677 (acid C=O spectra), and 1641 (amide C=O spectra) cm^−1^. In addition, the peaks observed at 1596 and 1503 cm^−1^ indicated the presence of quinoid and benzenoid rings. On the other hand, EPI showed typical asymmetric and symmetric carbonyl group absorption bands at 1715 and 1675 cm^−1^. Moreover, the absorption band of the carboxylic acid group completely disappeared, which indicated the complete conversion of EPAA to EPI.

### 3.3. Structural and Morphological Characterization of EPIS, EPIP, and EPIG

Various characterization tools were used for the determination of structure and morphology of the as-prepared materials. 

#### 3.3.1. X-ray Diffraction (XRD)

The XRD patterns clearly indicates that EPIS, EPIP, and EPIG had a crystalline nature as shown in Figure 4a. It can be seen that as compared to EPI and EPIS, EPIG exhibited four additional peaks at 2θ of 38.01°, 43.96°, 64.50°, and 77.42°. All these peaks were attributed to the standard Bragg reflection from (111), (200), (220), and (311) of the crystallographic planes of the face-centered cubic lattice. The intense peak at 38.1 indicated that the growth of gold was fixed in (111) direction. Likewise, the EPIP curve showed the crystal-plane peaks of (111), (200), and (220) of platinum metal at 2θ of 39.5°, 44.6°, and 64.9°. Thus, we may infer that the composition of the respective material was as expected, and no impurities were found in the XRD pattern [57,65,66].

#### 3.3.2. X-ray Photoelectron Spectroscopy (XPS)

The XPS analysis was employed to determine the composition of all the given samples as shown in Figure 4b. The XPS spectra of EPIS, EPIP, and EPIG clearly showed peaks for C, N, and O atoms, while the absence of any other peak indicated the purity of the given materials. The binding energy peak at 287.82 eV arose from C 1 s, while the peak at 399.8 eV corresponded to N 1 s. This peak was developed due to interactions with the N atom and respective nanoparticles. In addition, for EPIS the binding energy of Ag 3d_5/2_ Ag 3d_3/2_ was found to be 371.5 eV and 365.6 eV, respectively, which were comparable to the values reported in the literature for bulk Ag crystals (368 eV, 374 eV) [67]. Similarly, in case of EPIP, peaks appeared at 71.3 eV and 74.6 eV for Pt 4f_7/2_ and 4f_5/2_, respectively, corresponding to the zero-oxidation state [68]. Moreover, the EPIG XPS signal for Au 4f_7/2_ and Au 4f_5/2_ was observed as doublets at 81.5 and 85.3 eV, respectively, which clearly indicates the reduction of Au^3+^ into Au^0^, as the binding energy of Au 4f_7/2_ for Au^+3^ is 86.5 eV [54].

#### 3.3.3. Transmission Electron Microscopy (TEM)

Figure 4c–e presents the TEM images of EPIS, EPIP, and EPIG, respectively, under high magnification (200 K). The reduced metal NPs developed on the EPI surface were observed as black spots as shown in the Figure The average particle size of EPIS, EPIP, and EPIG was found to be ≈22 nm, ≈68 nm, and ≈20 ± 2 nm in diameter, respectively, as seen in Figure 4c–e.

### 3.4. Redox Capability of Materials Measured by Electrochemical CV Studies

The cyclic voltammograms (CV) of all EPIs are shown in Figure 5. As no redox peak was observed for bare ITO electrodes, it was used as a control for 5a. The CV curve for EPI 5b showed a relatively small peak of a redox current at 100 A cm^−2^_._ However, the doping of EPI with respective metal nanoparticles resulted in an increase in current density. For example, the current density for EPIS 5c was 225 A cm^−2^, which was 2.3 times higher than that for EPI, while the response for EPIP 5d increased by four times with a current density of 400 A cm^−2^. Subsequently, the highest response was observed for EPIG with a current density of about 600 A cm^−2^, as shown in Figure 5e. Thus, the redox capability of the as-prepared ITO electrode coated with electroactive materials was decreased in the following order: EPIG > EPIP > EPIS > EPI.

### 3.5. Kinetic Parameters Study of CPE Modified with EPIG

To study the nature and kinetics of the electrochemical reaction, the effect of the scan rate on CV was determined on a CPE modified with EPIG was used in 0.1 M PBS (pH = 7) towards the oxidation of 50 μM UA as evident in Figure 6a. It can be seen that with an increasing scan rate, the current density increased from 3 µA cm^−2^ to 8.6 µA cm^−2^. Furthermore, the peak potential shifted towards the positive side with the increasing scan rate. Figure 6b showed a linear relationship between the peak current (I_p_) and square root of the scan rates (ν_1/2_), and the regression equation was I_p_ (μA) = 11.147 _ν1/2_(V_S__−1_)_1/2_ + 0.562 with a correlation coefficient of R^2^ = 0.9958. Thus, it indicated that the UA oxidation process proceeds via a diffusion-controlled electron transfer reaction, which is generally observed in systems when the kinetics are fast. Additionally, to investigate the mechanism of UA electrocatalysis on an EPIG, a graph was plotted between the peak potential (E_p_) and logν as shown in Figure 6c. From the curve, it can be seen that E_p_ was proportional to logν by the following linear equation E_p_ = 0.1112 log (ν) + 0.5346, R^2^ = 0.97548, which indicates that the reaction is irreversible. Hence, an anodic peak potential can be evidenced by [69]
(1)$$\mathrm{Ep}=\frac{2.303\mathrm{RT}}{2\left(1-\alpha \right)\mathrm{naF}}\cdot \log\vartheta +k$$
where R and F are the universal gas constant and Faraday constant, respectively; α is the electron transfer coefficient; n_a_ is the number of electrons transferred; and T is the room temperature. The Tafel slope value calculated from Figure 6c was 0.222 V/decade, which shows that one electron transferred in a rate-determining step by assuming α = 0.61.

### 3.6. Electrochemical Sensing of UA

CV and amperometric studies were used to determine the electrocatalytic activity of the respective material for the detection of UA. Figure 7a showed the CV response corresponding to a bare CPE and a CPE modified with EPI, EPIS, EPIP, and EPIP in 0.1 M PBS (pH = 7) at 350 M UA concentration with scan rate of 50 mV s^−1^. As observed, the bare CPE 7a showed the smallest response with a current density of 12.73 uA cm^−2^ at a more positive potential of 459 mV, while, the current density for the CPE modified with EPI, EPIS, EPIP, and EPIG was found to be 28.10, 29.67, 32.68, and 37.39 uA cm^−2^, respectively. It may be conclude that UA sensitivity decreases in the following order EPIG > EPIP > EPIS > EPI.

The quantitative detection of UA with bare CPE and the EPI, EPIS, EPIP, and EPIG modified CPE was investigated by chronoamperometric measurements as shown in Figure 7b. The response was obtained by plotting a current–time (i–t) curve for 350 M UA at 340 mV in 30 mL of PBS solution (pH = 7) under constant stirring. It is noteworthy that the current density increases after each addition of UA. Further, a rapid and sensitive response was observed in the case of a CPE modified with EPIG, the sensitivity of which was calculated to be 0.153 A M^−1^, which is 1.84-fold, 3.56-fold, 4.5-fold, and 6.12-fold higher than those of a CPE modified with EPIP, EPIS, EPI, and a bare CPE, respectively, under the same analytical conditions as shown in Table 1.

At a constant potential of 340 mV and a lower concentration (0–50 µM), all the electrochemical sensors showed a linear relationship between current response and UA, as shown in Figure 7c. However, the response curve for EPIG was steep among all electrodes with a high sensitivity (0.153 A M^−1^) and having a good correlation coefficient (R^2^ = 0.9951). On the other hand, at a higher concentration of UA, the relationship between current density, and UA concentration was determined by the following equation [70]:j = j_max_[1 − exp(−c/b)](2)

(c = concentration of UA, j = oxidation current, b = empirical coefficient related to the steepness of the j, c transient curve and j_max_ = current maxima at higher UA concentration). Based on the data, the previous equation yielded j_max_ = 15.33 A cm^−2^, b = 92.59 M, and a correlation coefficient of R^2^ = 0.9968. However, it is not easy to apply the above equation for a current vs concentration plot. Thus, we generated a linear relationship by taking the log of the abscissa of the calibration curve. So, the sensitivity values calculated for a CPE modified with an EPIG, EPIP, EPIS, EPI, and a bare CPE was 0.153, 8.3 × 10^−2^, 4.7 × 10^−2^, 3.4 × 10^−2^, and 2.5 × 10^−2^ A M^−1^, respectively. In addition, the current density increased with the successive addition of UA, while the steady state was achieved after 2 s. These results indicated that, an EPIG is a highly sensitive sensor with a rapid response, wide linear dynamic range (LDR) and low limit of detection (LOD) (Table 1).

### 3.7. Differential Pulse Voltammetric Responses for Tertiary Mixtures of AA, UA, and DA

Differential pulse voltammograms (DPVs) were used [80] to determine the selectivity of designed electrochemical sensors. The experiment was carried out using a tertiary mixture of 10 M of DA, 10 M of AA, and 10 M of UA for a bare CPE, a CPE modified with an EPI, EPIS, EPIP, and EPIG in 0.1 M PBS (pH = 7) in the range of −0.3 to 0.8 V, as shown in Figure 8. It can be seen that the bare CPE 8a showed poor selectivity for a tertiary mixture of AA/DA/UA. However, an improved and clearly distinguished signal can be seen for a CPE modified with EPI 8b, EPIS 8c, EPIP 8d, and EPIG 8e. The overall current value of a CPE modified with metal NPs increased as compared to a EPI due to electrocatalysis. The separation potential was also significantly improved. It showed good separation with three oxidation peaks around 10–50 mV, 160–180 mV, and 300–340 mV, which corresponded to the anodic potentials of AA, DA, and UA respectively. Due to the excellent electrocatalysis of EPIG as caused by Au NPs, the overall sensing current value was increased, depicting excellent selectivity, good resolution and prominent voltammetric peaks.

## 4. Conclusions

In conclusion, we fabricated stable and sensitive UA sensors based on EPI decorated with reduced nanoparticles of Ag, Pt, and Au, i.e., EPIS, EPIP, and EPIG, respectively. The characterization of the as-prepared materials via XRD, XPS, CV, and TEM demonstrated the metallic nanoparticles were successfully reduced upon the electroactive segment of the EPI. The redox capability of the UA sensors based on CV studies decreased in the following order EPIG > EPIP > EPIS > EPI > ITO, which implied that EPIG may reveal the highest sensitivity. The amperometric data showed that the UA sensor EPIG exhibited higher sensitivity (0.153 A^.^M^−1^ UA), low LOD (0.78 µM,) and wide LDA (5–50 µM UA) at a lower oxidation potential of 310 mV. Similarly, the DPV results clearly indicated the excellent selectivity of EPIG as compared to other constructed sensors. Hence, the EPIG electrochemical sensor developed for UA detection showed reliable results. Considering the good selectivity, simple preparation, low cost, and selectivity of the modified sensor of UA the practical application value of this system will be later evaluated using real samples.

## Data Availability

No new data were created or analyzed in this study. Data sharing is not applicable to this article.

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
