# Peer review of "Comparative Studies of CPEs Modified with Distinctive Metal Nanoparticle-Decorated Electroactive Polyimide for the Detection of UA"

_polymers, 2021, doi:10.3390/polym13020252_

Round 1
Reviewer 1 Report
The manuscript "Comparative studies of carbon paste electrodes modified with electroactive polyimide/metal nanoparticles for the detection of uric acid " is very interesting and is well written. The abstract gives a concise summary of the manuscript. The results are adequate and well analysed. The conclusions and discussion should be improved, specially highlighting other important work published in literature (more relevant references should be introduced for example: please see below). Therefore, minor corrections should be amended before being accepted.
- C. Fernandez et al
Microchemical Journal 160, 105668, 2021
Electroanalytical determination of gallic acid in red and white wine samples using cobalt oxide nanoparticles-modified carbon-paste electrodes
https://doi.org/10.1016/j.microc.2020.105668
- C. Chikere et al
Nanomaterials, 10, 537, 1-25, 2020
Interaction between Amorphous Zirconia Nanoparticles and Graphite: Electrochemical Applications for Gallic Acid Sensing Using Carbon Paste Electrodes in Wine
https://doi.org/10.3390/nano10030537
- N.H. Faisal et al
Journal of physics: conference series [online], 1310: proceedings of 2018 Applied nanotechnology and nanoscience international conference (ANNIC 2018), 22-24 October 2018, Berlin, Germany, article ID 012008
Zinc oxide nanoparticles modified-carbon paste electrode used for the electrochemical determination of Gallic acid
https://doi.org/10.1088/1742-6596/1310/1/012008
Author Response
We have mentioned the results from research articles proposed by the reviewer in table I for the comparison of various parameters as reference 68, 69. We have mentioned the third articles as a reference 70 for the use of DPV for the determination of sensor selectivity.
Reviewer 2 Report
Author reported EPI-nano particles based electrochemical sensors for UA detection. The topic is old since the electrochemical UA sensor already commercialized. However, The synthesis of EPI-nano particles deserves to be studied.Therefore, I think this work can be accepted after modification.
Specific comments:
- The electrochemical sensor for UA detection is a very old topic. Authors should include more information in the Introduction section. Specifically, author should emphasize the purpose of design a new sensor.
- line 54, missing subscript.
- Citation should be included after the sentence " OAEP mostly reported for electrochemical sensing of ascorbic acid."
- Scheme 1 is not particularly required.
- For FTIR characterization, author forgot sign the peaks between 1000 to 1500 cm-1 of ACAT.
- Why TEM of EPIP has two different background color?
- Author also required adding some references for comparison in Table 1. I suggest following references may include: Talanta 180 (2018): 248; Arabian Journal for Science and Engineering 41.1 (2016): 135.
Author Response
1. Thankyou, the purpose of this sensor is to improve the various parameters of electrochemical sensing by doing a comparative study of reduced noble metal particles on Electroactive Polyimide (EPI) for Uric acid sensing, as mentioned in the last paragraph of our introduction section
2. thankyou, the correction was made.
3. Thankyou for your suggestion, the citation was added with correction
4. Thankyou for the suggestion, I have removed the scheme I.
5. Thankyou for correcting me, FTIR peaks for ACAT has been marked.
6. it may be due to instrument issue.
7. Thankyou for pointing out, various electrochemical sensing parameters from articles mentioned by the reviewer was added in table I.
Reviewer 3 Report
The results described deserve publication, but it is necessary to improve their presentation.
All pictures and plots have to be the same good quality (such as e.g. Fig. 3 and 8);
similarly, Scheme II needs an improvement to be well readable (perhaps it could be simplified).
Fig. 1 is trivial.
Author Response
- Thank you for your kind suggestion, Scheme II was revised and simplified.
- Thank you for pointing, fig 1 was revised.
Round 2
Reviewer 2 Report
The revised version can be accepted.